# Entropy, Free Energy, and Work of Restricted Boltzmann Machines

**DOI:** 10.3390/e22050538

**Published:** 2020-05-11

**Authors:** Sangchul Oh, Abdelkader Baggag, Hyunchul Nha

**Affiliations:** 1Qatar Environment and Energy Research Institute, Hamad Bin Khalifa University, Qatar Foundation, 5825 Doha, Qatar; 2Qatar Computing Research Institute, Hamad Bin Khalifa University, Qatar Foundation, 5825 Doha, Qatar; abaggag@hbku.edu.qa; 3Department of Physics, Texas A&M University at Qatar, Education City, 23874 Doha, Qatar; hyunchul.nha@qatar.tamu.edu

**Keywords:** restricted Boltzmann machines, entropy, subadditivity of entropy, Jarzynski equality, machine learning

## Abstract

A restricted Boltzmann machine is a generative probabilistic graphic network. A probability of finding the network in a certain configuration is given by the Boltzmann distribution. Given training data, its learning is done by optimizing the parameters of the energy function of the network. In this paper, we analyze the training process of the restricted Boltzmann machine in the context of statistical physics. As an illustration, for small size bar-and-stripe patterns, we calculate thermodynamic quantities such as entropy, free energy, and internal energy as a function of the training epoch. We demonstrate the growth of the correlation between the visible and hidden layers via the subadditivity of entropies as the training proceeds. Using the Monte-Carlo simulation of trajectories of the visible and hidden vectors in the configuration space, we also calculate the distribution of the work done on the restricted Boltzmann machine by switching the parameters of the energy function. We discuss the Jarzynski equality which connects the path average of the exponential function of the work and the difference in free energies before and after training.

## 1. Introduction

A restricted Boltzmann machine (RBM) [1] is a generative probabilistic neural network. RBMs and general Boltzmann machines are described by a probability distribution with parameters, i.e., the Boltzmann distribution. An RBM is an undirected Markov random field and is considered a basic building block of deep neural networks. RBMs have been applied widely, for example, in dimensionality reduction, classification, feature learning, pattern recognition, topic modeling, and so on [2,3,4].

As its name implies, the RBM is closely connected to physics and they share some important concepts such as entropy, free energy, and so forth [5]. Recently, RBMs have gained renewed attention in physics since Carleo and Troyer [6] showed that a quantum many-body state could be efficiently represented by the RBM. Gabré et al. and Tramel et al. [7] employed the Thouless–Anderson–Palmer mean-field approximation, used for a spin glass problem, to replace the Gibbs sampling of contrastive-divergence training. Amin et al. [8] proposed a quantum Boltzmann machine based on the quantum Boltzmann distribution of a quantum Hamiltonian. More interestingly, there is a deep connection between the Boltzmann machine and tensor networks of quantum many-body systems [9,10,11,12,13]. Xia and Kais combined the restricted Boltzmann machine and quantum computing algorithms to calculate the electronic energy of small molecules [14].

While the working principles of RBMs have been well established, it may still be needed to understand the RBM better for further applications. In this paper, we investigate the RBM from the perspective of statistical physics. As an illustration, for bar-and-stripe pattern data, the thermodynamic quantities such as the entropy, the internal energy, the free energy, and the work are calculated as a function of the epoch. Since the RBM is a bipartite system composed of visible and hidden layers, it may be interesting, and informative, to see how the correlation between the two layers grows as the training goes on. We show that the total entropy of the RBM is always less than the sum of the entropies of visible and hidden layers, except at the initial time when the training begins. This is the so-called subadditivity of entropy, indicating that the visible layer becomes correlated with the hidden layer as the training proceeds. The training of the RBM is to adjust the parameters of the energy function, which can be considered as the work done on the RBM, from a thermodynamic point of view. Using the Monte-Carlo simulation of the trajectories of the visible and hidden vectors in the configuration space, we calculate the work of a single trajectory and the statistics of the work over the ensemble of trajectories. We also examine the Jarzynski equality that connects the ensemble of the work done on the RBM and the difference in free energies before and after the training of the RBM.

The paper is organized as follows. In Section 2, a detailed analysis of the RBM from the statistical physics point of view is described. In Section 3, we present the summary of the result together with discussions.

## 2. Statistical Physics of Restricted Boltzmann Machines

### 2.1. Restricted Boltzmann Machines

Let us start with a brief introduction of the RBM [1,2,3]. As shown in Figure 1, the RBM is composed of two layers; the visible layer and the hidden layer. Possible configurations of the visible and hidden layers are represented by the random binary vectors, $v=\left(v_{1},\cdots ,v_{N}\right)\in {\left\{0,1\right\}}^{N}$ and $h=\left(h_{1},\cdots ,h_{M}\right)\in {\left\{0,1\right\}}^{M}$, respectively. The interaction between the visible and hidden layers is given by the so-called weight matrix $w\in R^{N}\times R^{M}$, where the weight $w_{ij}$ is the connection strength between a visible unit $v_{i}$ and a hidden unit $h_{j}$. The biases $b_{i}\in R.$ and $c_{j}\in R$ are applied to visible unit *i* and hidden unit *j*, respectively. Given random vectors v and h, the energy function of the RBM is written as an Ising-type Hamiltonian.
(1)$$E\left(v,h;\theta \right)=-\sum_{i=1}^{N}\sum_{j=1}^{M}w_{ij}v_{i}h_{j}-\sum_{i=1}^{N}b_{i}v_{i}-\sum_{i=1}^{M}c_{i}h_{i}\;,$$
where the set of model parameters is denoted by $\theta \equiv \{w_{ij},b_{i},c_{j}\}$. The joint probability of finding v and h of the RBM in a particular state is given by the Boltzmann distribution
(2)$$p\left(v,h;\theta \right)=\frac{e^{-E(v,h;\theta )}}{Z(\theta )},$$
where the partition function, $Z\left(\theta \right)\equiv \sum_{v,h}e^{-E(v,h;\theta )}$, is the sum over all possible configurations. The marginal probabilities p(v;θ) and p(h;θ) for visible and hidden layers are obtained by summing up the hidden or visible variables, respectively,
(3)$$\begin{array}{cc} p(v;\theta )\; & =\sum_{h}p\left(v,h;\theta \right)=\frac{1}{Z(\theta )}\sum_{h}e^{-E(v,h;\theta )}\;, \end{array}$$
(4)$$\begin{array}{cc} p(h;\theta )\; & =\sum_{v}p\left(v,h;\theta \right)=\frac{1}{Z(\theta )}\sum_{v}e^{-E(v,h;\theta )}\;. \end{array}$$

The training of the RBM is to adjust the model parameter θ such that the marginal probability of the visible layer p(v;θ) becomes as close as possible to the unknown probability $p_{\mathrm{data}}\left(v\right)$ that generate the training data. Given identically and independently sampled training data $D\in \{v^{\left(1\right)},\ldots ,v^{\left(D\right)}\}$, the optimal model parameters θ can be obtained by maximizing the likelihood function of the parameters, $L\left(\theta |D\right)=\prod_{i=1}^{D}p\left(v^{\left(i\right)};\theta \right)$, or equivalently by maximizing the log-likelihood function $\ln L\left(\theta |D\right)=\sum_{i=1}^{D}\ln p\left(v^{\left(i\right)};\theta \right)$. Maximizing the likelihood function is equivalent to minimizing the Kullback–Leibler divergence or the relative entropy of p(v;θ) from q(v) [15,16]
(5)$$D_{\mathrm{KL}}\left(q\;||\;p\right)=\sum_{v}q\left(v\right)\ln\frac{q(v)}{p(v;\theta )}\;,$$
where q(v) is an unknown probability that generates the training data, $q\left(v\right)=p_{\mathrm{data}}\left(v\right)$. Another method of monitoring the progress of training is the cross-entropy cost between the input visible vector $v^{\left(i\right)}$ and a reconstructed visible vector ${\bar{v}}^{\left(i\right)}$ of the RBM,
(6)$$C=-\frac{1}{D}\sum_{i\in D}\left[v^{\left(i\right)}\ln{\bar{v}}^{\left(i\right)}+\left(1-v^{\left(i\right)}\right)\ln\left(1-{\bar{v}}^{\left(i\right)}\right)\right]\;.$$

The stochastic gradient ascent method for the log-likelihood function is used to train the RBM. Estimating the log-likelihood function requires the Monte-Carlo sampling for the model probability distribution. Well-known sampling methods are the contrastive-divergence, denoted by CD-*k*, and the persistent contrastive divergence PCD-*k*. For details of the RBM algorithm, please see References [2,3,4]. Here, we employ the CD-*k* method.

### 2.2. Free Energy, Entropy, and Internal Energy

From a physics point of view, the RBM is a finite classical system composed of two subsystems, similar to an Ising spin system. The training of the RBM is considered the driving of the system from an initial equilibrium state to the target equilibrium state by switching the model parameters. It may be interesting to see how thermodynamic quantities such as free energy, entropy, internal energy, and work change as the training progresses.

It is straightforward to write down various thermodynamic quantities for the total system. The free energy *F* is given by the logarithm of the partition function *Z*,
(7)F(θ)=−lnZ(θ).

The internal energy *U* is given by the expectation value of the energy function E(v,h;θ)
(8)$$U\left(\theta \right)=\sum_{v,h}E\left(v,h;\theta \right)p\left(v,h;\theta \right)\;.$$

The entropy *S* of the total system comprising the hidden and visible layers is given by
(9)$$S\left(\theta \right)=-\sum_{v,h}p\left(v,h;\theta \right)\ln p\left(v,h;\theta \right)\;.$$

Here, the convention of 0ln0=0 is employed if p(v,h)=0 [17]. The free energy (Equation 7) is related to the difference between the internal energy (Equation 9) and the entropy (Equation 10)
(10)F=U−TS,
where *T* is set to 1.

Generally, it is very challenging to calculate the thermodynamic quantities, even numerically. The number of possible configurations of *N* visible units and *M* hidden units grow exponentially as $2^{N+M}$. Here, for a feasible benchmark test, the 2×2 bar-and-stripe data are considered [18,19]. Figure 2 shows the 6 possible 2×2 bar-and-stripe patterns out of 16 possible configurations, which will be used as the training data in this work. We take the sizes of the visible and the hidden layers as N=4 and M=6, respectively. One may take a larger size of hidden layers, i.e., *M* = 8 or 10, but it does not make an appreciable difference in our results. *M* = 6 is not a choice of magic number but was used as an example since we were rather limited in our capacity of numerical computation. In order to understand better how the RBM is trained, the thermodynamic quantities are calculated numerically for this small benchmark system.

Figure 3 shows how the weight $w_{ij}$, the bias $b_{i}$ on the visible unit *i* and the bias $c_{j}$ on the hidden unit *j* change as the training goes on. The weights $w_{ij}$ are clustered into three classes. The evolution of the bias $b_{i}$ on the visible layer is somewhat different from that of the bias $c_{j}$ on the hidden layer. The change in $c_{i}$ is larger than that in $b_{i}$. Figure 4 shows the change in the marginal probabilities p(v) of the visible layer and p(h) of the hidden layer before and after training. Note that the marginal probability p(v) after training is not distributed exclusively over six possible outcomes corresponding to the training data set in Figure 2.

Typically, the progress of learning of the RBM is monitored by the loss function. Here, the Kullback–Leibler divergence, Equation (Equation 5), and the reconstructed cross entropy, Equation (Equation 6), are used. Figure 5 plots the reconstructed cross entropy *C*, the Kullback–Leibler divergence $D_{\mathrm{KL}}$, the entropy *S*, the free energy *F*, and the internal energy *U* as a function of the epoch. As shown in Figure 5a, it is interesting to see that even after a large number of epochs ∼10,000, the cost function *C* continues approaching zero while the entropy *S* and the Kullback–Leibler divergence $D_{\mathrm{KL}}$ become steady. On the other hand, the free energy *F* continues decreasing together with the internal energy *U*, as depicted in Figure 5b. The Kullback–Leibler divergence is a well-known indicator of the performance of RBMs. Then, our result implies that the entropy may be another good indicator to monitor the progress of the RBM while other thermodynamic quantities may be not.

In addition to the thermodynamic quantities of the total system of the RBM, Equations (Equation 7)–(Equation 9), it is interesting to see how the two subsystems of the RBM evolve. Since the RBM has no intra-layer connection, the correlation between the visible layer and the hidden layer may increase as the training proceeds. The correlation between the visible layer and the hidden layer can be measured by the difference between the total entropy and the sum of the entropies of the two subsystems. The entropies of the visible and hidden layers are given by
(11)$$\begin{array}{cc} S_{V} & =-\sum_{v}p\left(v;\theta \right)\ln p\left(v;\theta \right)\;, \end{array}$$
(12)$$\begin{array}{cc} S_{H} & =-\sum_{h}p\left(h;\theta \right)\ln p\left(h;\theta \right)\;. \end{array}$$

The entropy $S_{V}$ of the visible layer is closely related to the Kullback–Leibler divergence of p(v;θ) to an unknown probability q(v) which produces the data. Equation (Equation 5) is expanded as
(13)$$\begin{array}{c} D_{\mathrm{KL}}\left(q\;||\;p\right)=\sum_{v}q\left(v\right)\ln q\left(v\right)-\sum_{v}q\left(v\right)\ln p\left(v;\theta \right)\;. \end{array}$$

The second term $-\sum_{v}q\left(v\right)\ln p\left(v;\theta \right)$ depends on the parameter θ. As the training proceeds, p(v;θ) becomes close to q(v) so the behavior of the second term is very similar to that of the entropy $S_{V}$ of the visible layer. If the training is perfect, we have q(v)=p(v;θ) that leads to $D_{\mathrm{KL}}\left(q\;||\;p\right)=0$ while $S_{V}$ remains nonzero.

The difference between the total entropy and the sum of the entropies of subsystems is written as
(14)$$S-\left(S_{V}+S_{H}\right)=\sum_{v,h}p\left(v,h\right)\ln\left[\frac{p(v)\;p(h)}{p(v,h)}\right]\;.$$

Equation (Equation 14) tells us that if the visible random vector v and the hidden random vector h are independent, i.e., p(v,h;θ)=p(v;θ)p(h;θ), then the entropy *S* of the total system is the sum of the entropies of subsystems. In general, the entropy *S* of the total system is always less than or equal to the sum of the entropy of the visible layer, $S_{V}$, and the entropy of the hidden layer, $S_{H}$,[20]],
(15)$$S\leq S_{V}+S_{H}\;.$$

This is called the subadditivity of entropy, one of the basic properties of the Shannon entropy, which is also valid for the von Neumann entropy [17,21]. This property can be proved using the log inequality, −lnx≥−x+1. In another way, Equation (Equation 15) may be proved by using the log-sum inequality, which states that for the two sets of nonnegative numbers, $a_{1},\ldots ,a_{n}$ and $b_{1},\ldots ,b_{n}$,
(16)$$\sum_{i}a_{i}\log\frac{a_{i}}{b_{i}}\geq \left(\sum_{i}a_{i}\right)\log\frac{\left(\sum_{i}a_{i}\right)}{\left(\sum_{i}b_{i}\right)}.$$

In other words, Equation (Equation 14) can be regarded as the negative of the relative entropy or Kullback–Leibler divergence of the joint probability p(v,h) to the product probability p(v)·p(h),
(17)$$I\left(p(v,h)\;||\;p(v)p(h)\right)=\sum_{v,h}p\left(v,h\right)\log\left[\frac{p(v,h)}{p(v)p(h)}\right]\;.$$

For the 2×2 bar-and-stripe pattern, the entropies of visible and hidden layers, $S_{V},S_{H}$ are calculated numerically. Figure 6 plots the entropies, $S_{V},S_{H}$, *S*, and the Kullback–Leibler divergence $D_{\mathrm{KL}}\left(q\;||\;p\right)$ as a function of the epoch. Figure 6a shows that the Kullback–Leibler divergence, $D_{\mathrm{KL}}\left(q\;||\;p\right)$ becomes saturated, though above zero, as the training proceeds. Similarly, the entropy $S_{V}$ of the visible layer is saturated. This implies that the entropy of the visible layer, as well as the total entropy shown in Figure 5, can be a better indicator of learning than the reconstructed cross entropy *C*, Equation (Equation 6). The same can also be said about the entropy of the hidden layer, $S_{H}$. If some information measures such as entropy and the Kullback–Leibler divergence become steady, one may presume the training has been done.

The difference between the total entropy and the sum of the entropies of the two subsystems, $S-(S_{V}+S_{H})$, becomes less than 0, as shown in Figure 6b. Thus, it demonstrates the subadditivity of entropy, i.e., the correlation between the visible and the hidden layer as the training proceeds. As it is saturated just as the total entropy and the entropies of the visible and hidden layers after a large number of epochs, the correlation between the visible layer and the hidden layer can also be a good quantifier of the RBM progress.

### 2.3. Work, Free Energy, and Jarzynski Equality

The training of the RBM may be viewed as driving a finite classical spin system from an initial equilibrium state to a final equilibrium state by changing the system parameters θ slowly. If the parameters θ are switched infinitely slowly, the classical system remains in a quasi-static equilibrium. In this case, the total work done on the systems is equal to the Helmholtz free energy difference between the before-training and the after-training, $W_{\infty }=F_{1}-F_{0}\;.$ For switching θ at a finite rate, the system may not evolve immediately to an equilibrium state, the work done on the system depends on a specific path of the system in the configuration space. Jarzynski [22,23] proved that for any switching rate, the free energy difference ΔF is related to the average of the exponential function of the amount of work *W* over the paths
(18)$${\langle e^{-W}\rangle }_{\mathrm{path}}=e^{-\Delta F}\;.$$

The RBM is trained by changing the parameters θ through a sequence $\left\{\theta_{0},\theta_{1},\ldots ,\theta_{\tau }\right\}$, as shown in Figure 3. To calculate the work done during the training, we perform the Monte-Carlo simulation of the trajectory of a state (v,h) of the RBM in configuration space. From the initial configuration, $\left(v_{0},h_{0}\right)$ which is sampled from the initial Boltzmann distribution, Equation (Equation 2), the trajectory $\left(v_{0},h_{0}\right)\to \left(v_{1},h_{1}\right)\to \cdots \to \left(v_{\tau },h_{\tau }\right)$ is obtained using the Metropolis–Hastings algorithm of the Markov chain Monte-Carlo method [24,25]. Assuming the evolution is Markovian, the probability of taking a specific trajectory is the product of the transition probabilities at each step,
(19)$$\begin{array}{cc} p\left(v_{0},h_{0}\overset{\theta_{1}}{⟶}v_{1},h_{1}\right)\;p\left(v_{1},h_{1}\overset{\theta_{2}}{⟶}v_{2},h_{2}\right) & \\ \ldots p(v_{\tau -1},h_{\tau -1}\overset{\theta_{\tau }}{⟶}v_{\tau },h_{\tau }) & \;. \end{array}$$

The transition $\left(v,h\right)\to \left(v^{\prime },h^{\prime }\right)$ can be implemented by the Metropolis–Hastings algorithm based on the detailed balance condition for the fixed parameter θ,
(20)$$\begin{array}{c} \frac{p(v,h\overset{\theta }{⟶}v^{\prime },h^{\prime })}{p(v,h\overset{\theta }{⟵}v^{\prime },h^{\prime })}=\frac{e^{-E(v^{\prime },h^{\prime };\theta )}}{e^{-E(v,h;\theta )}}\;. \end{array}$$

The work done on the RBM at epoch *i* may be given by
(21)$$\delta W_{i}=E\left(v_{i},h_{i};\theta_{i+1}\right)-E\left(v_{i},h_{i};\theta_{i}\right)\;.$$

The total work $W=\sum \delta W_{i}$ performed on the system is written as [26]
(22)$$\begin{array}{cc} W & =\sum_{i=0}^{\tau -1}\left[E\left(v_{i},h_{i};\theta_{i+1}\right)-E\left(v_{i},h_{i};\theta_{i}\right)\right]\;. \end{array}$$

Given the sequence of the model parameter $\left\{\theta_{0},\theta_{1},\ldots ,\theta_{\tau }\right\}$, the Markov evolution of the visible and hidden vectors $\left(v,h\right)\in {\left\{0,1\right\}}^{N+M}$ may be considered the discrete random walk. Random walkers move to the points with low energy in configuration space. Figure 7 shows the heat map of energy function E(v,h;θ) of the RBM for the 2×2 bar-and-stripe pattern after training. One can see the energy function has deep levels at the visible vectors corresponding to the bar-and-stripe patterns of the training data set in Figure 2, representing a high probability of generating the trained patterns. Furthermore, note that the energy function has many local minima. Figure 8 plots a few Monte-Carlo trajectories of the visible vector v as a function of the epoch. Before training, the visible vector v is distributed over all possible configurations, represented by the number (0,⋯,15). As the training progresses, the visible vector v becomes trapped into one of the six possible outcomes (0,3,5,10,12,15).

In order to examine the relation between work done on the RBM during the training and the free energy difference, the Monte-Carlo simulation is performed to calculate the average of the work over paths generated by the Metropolis–Hastings algorithm of the Markov chain Monte-Carlo method. Each path starts from an initial state sampled from the uniform distribution over the configuration space, as shown in Figure 4a. Since the work done on the system depends on the path, the distribution of the work is calculated by generating many trajectories. Figure 9 shows the distribution of the work over 50000 paths at 5000 training epochs. The Monte-Carlo average of the work is 〈W〉≈−5.481, and its standard deviation is $\sigma_{W}\approx 3.358$. The distribution of the work generated by the Monte-Carlo simulation is well fitted to the Gaussian distribution, as depicted by the red curve in Figure 9. This agrees with the statement in Reference [23] that for the slow switching of the model parameters the probability distribution of work is approximated to the Gaussian.

We perform the Monte-Carlo calculation of the exponential average of work, ${\langle e^{-W}\rangle }_{\mathrm{path}}$ to check the Jarzynski equality, Equation (Equation 18). The free energy difference can be estimated as
(23)$$e^{-\Delta F}={\langle e^{-W}\rangle }_{\mathrm{path}}\approx \frac{1}{N_{\mathrm{mc}}}\sum_{n=1}^{N_{\mathrm{mc}}}e^{-W_{n}}\;,$$
where $N_{\mathrm{mc}}$ is the number of the Monte-Carlo samplings. At a small epoch number, the Monte-Carlo estimated value of the free energy difference is close to ΔF calculated from the partition function. However, this Monte-Carlo calculation gives rise to the poor estimation of the free energy difference if the epoch is greater than 5000. This numerical errors can be explained by the fact that the exponential average of the work is dominated by rare realization [27,28,29,30,31]. As shown in Figure 9, the distribution of work is given by the Gaussian distribution ρ(W) with the mean 〈W〉 and the standard deviation $\sigma_{W}$. If the standard deviation $\sigma_{W}$ becomes larger, the peak position of $\rho \left(W\right)e^{-W}$ moves to the long tail of the Gaussian distribution. So the main contribution of the integration of $\langle e^{-W}\rangle$ comes from the rare realizations. Figure 10 shows that the standard deviation $\sigma_{W}$ grows with the epoch, so the error of the Monte-Carlo estimation of the exponential average of the work grows quickly.

If $\sigma_{W}^{2}\ll k_{B}T$, the free energy is related to the average of work and its variance as
(24)$$\Delta F={\langle W\rangle }_{\mathrm{path}}-\frac{\sigma_{W}^{2}}{2k_{B}T}\;.$$

Here, the case is the opposite, the spread of the value of work is large, i.e., $\sigma_{W}^{2}\gg k_{B}T$(=1), so the central limit theorem does not work and the above equation can not be applied [32]. Figure 10 shows how the average of work, ${\langle W\rangle }_{\mathrm{path}}$, over the Markov chain Monte-Carlo paths changes as a function of the epoch. The standard deviation of the Gaussian distribution of the work also grows as a function of the training epoch. The free energy difference between before-training and after-training is called the reversible work $W_{r}=\Delta F$. The difference between the actual work and the reversible work is called the dissipative work, $W_{d}=W-W_{r}$ [26]. As depicted in Figure 10, the magnitude of the dissipative work grows with the training epoch.

## 3. Summary

In summary, we analyzed the training process of the RBM in the context of statistical physics. In addition to the typical loss function, i.e., the reconstructed cross entropy, the thermodynamic quantities such as free energy *F*, internal energy *U*, and entropy *S* were calculated as a function of the epoch. While the free energy and the internal energy decrease rather indefinitely with epochs, the total entropy and the entropies of the visible and the hidden layers become saturated together with the Kullback–Leibler divergence after a sufficient number of epochs. This result suggests that the entropy of the system may be a good indicator of the RBM progress along with the Kullback–Leibler divergence. It seems worth investigating the entropy for other larger data sets, for example, MNIST handwritten digits [33], in future works.

We have further demonstrated the subadditivity of the entropy, i.e., the entropy of the total system is less than the sum of the entropies of the two layers. This manifested the correlation between the visible and hidden layers growing with the training progress. Just as the entropies are well saturated together with the Kullback–Leibler divergence, so is the correlation that is determined by the total and the local entropies. In this sense, the correlation between the visible and the hidden layer may become another good indicator of the RBM performance.

We also investigated the work done on the RBM by switching the parameters of the energy function. The trajectories of the visible and hidden vectors in the configuration space were generated using the Markov chain Monte-Carlo simulation. The distribution of the work follows the Gaussian distribution and its standard deviation grows with the training epochs. We discussed the Jarzynski equality, which connects the free energy difference and the average of the exponential function of the work over the trajectories. We note that, in addition to the Jarzynski equality, the Crooks path-ensemble average method [34,35] with the forward and backward transformations could be also used to connect the free energy difference and the work. This is called the bidirectional estimator [36] in contrast to the unidirectional estimator such as the Jarzynski equality or the Hummer–Szabo method [37].

A more detailed analysis from a full thermodynamics or statistical physics point of view can bring us useful insights into the performance of the RBM. This course of study may enable us to come up with possible methods for a better performance of the RBM for many different applications in the long run. Therefore, it may be worthwhile to further pursue our study, e.g., a rigorous assessment of scaling behavior of thermodynamic quantities with respect to epochs as the sizes of the visible and hidden layers increase. We also expect that a similar analysis on a quantum Boltzmann machine can be valuable as well.

## Figures and Tables

**Figure 1 entropy-22-00538-f001:**
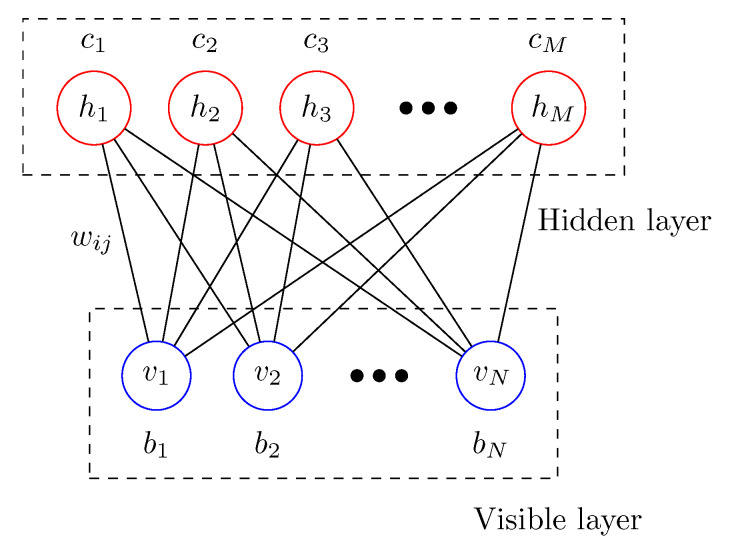
Graph structure of a restricted Boltzmann machine with the visible layer and the hidden layer.

**Figure 2 entropy-22-00538-f002:**
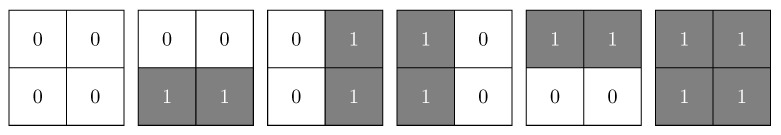
Six samples of 2×2 bar-and-stripe patterns used as the training data in this work. Each configuration is represented by a visible vector $v\in {\left\{0,1\right\}}^{2\times 2}$ or by a decimal number; (0,0,0,0)=0, (0,0,1,1)=3, (0,1,0,1)=5, (1,0,1,0)=10, (1,1,0,0)=12, (1,1,1,1)=15 in row-major ordering.

**Figure 3 entropy-22-00538-f003:**
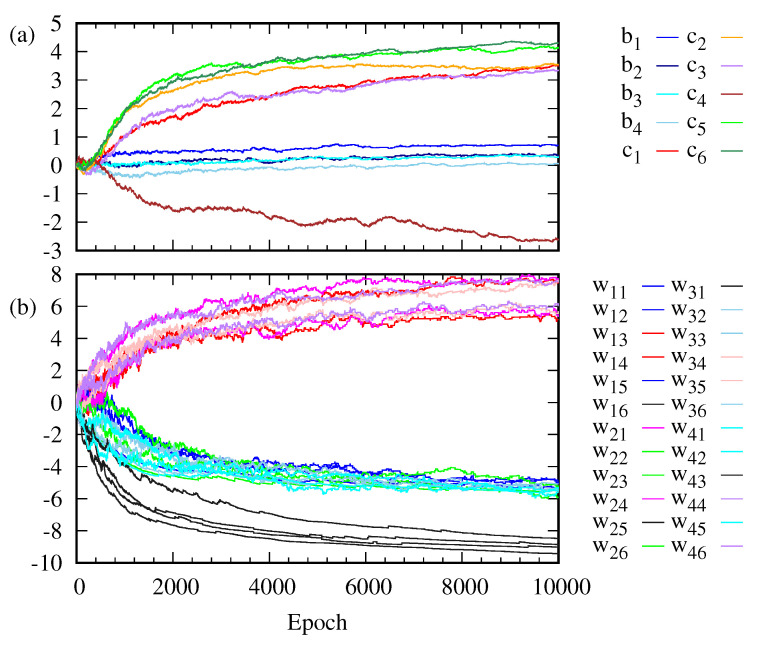
(**a**) Bias $b_{i}$ on the visible unit *i* and bias $c_{j}$ on the hidden unit *j* are plotted as a function of the epoch. (**b**) Weight $w_{ij}$ connecting the visible unit *i* and the hidden unit *j* are plotted as a function of the epoch.

**Figure 4 entropy-22-00538-f004:**
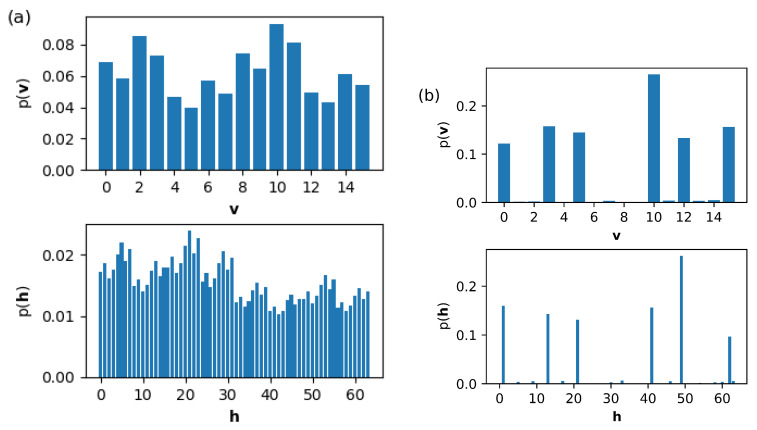
Marginal probabilities p(v) of visible layer and p(h) of hidden layer are plotted (**a**) before training and (**b**) after training. The binary vector v or h in the x-axis is represented by the decimal number as noted in the caption of Figure 2. The visible and the hidden layers have a total number of configurations given by $2^{4}=16$ and $2^{6}=64$, respectively. The learning rate is 0.15, the training epoch 20000, and k=100 in CD-*k*.

**Figure 5 entropy-22-00538-f005:**
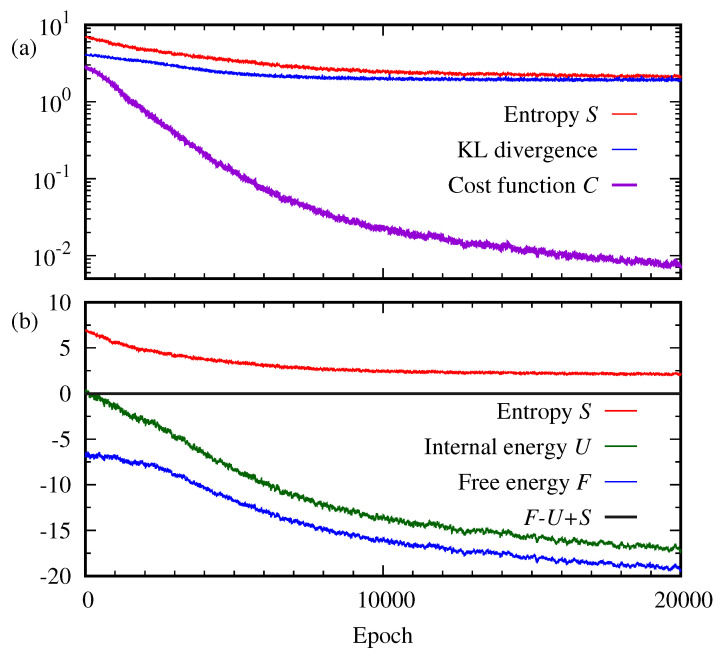
For 2×2 bar-and-stripe data, (**a**) cost function *C*, entropy *S*, and the Kullback–Leibler divergence $D_{\mathrm{KL}}\left(q\;||\;p\right)$ are plotted as a function of the epoch. (**b**) Free energy *F*, entropy *S*, and internal energy *U* of the RBM are calculated as a function of the epoch.

**Figure 6 entropy-22-00538-f006:**
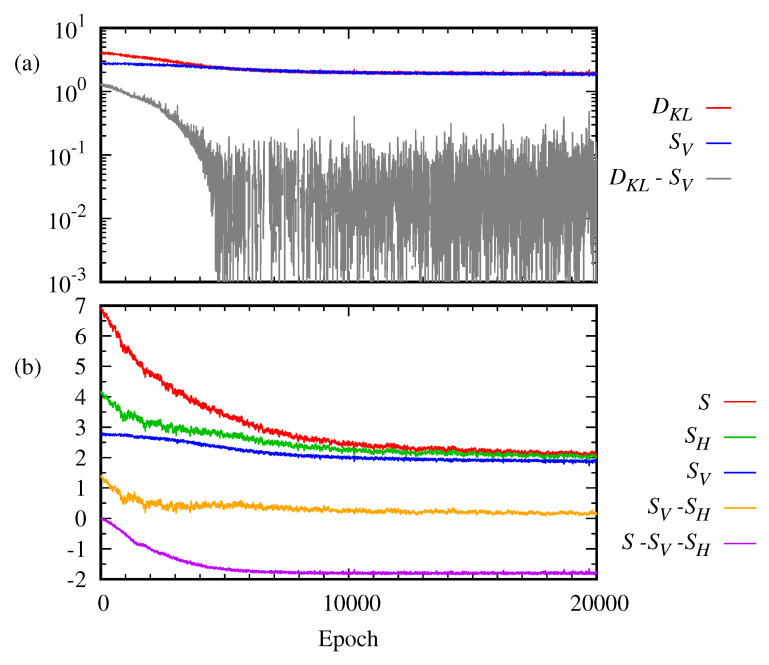
(**a**) Kullback–Leibler divergence $D_{\mathrm{KL}}\left(q\;||\;p\right)$, entropy $S_{V}$, and their difference are plotted as a function of the epoch. (**b**) Entropy *S* of the total system, entropy $S_{V}$ of the visible layer, entropy $S_{H}$ of the hidden layer, and the difference $S-S_{H}-S_{V}$ are plotted as a function of the epoch.

**Figure 7 entropy-22-00538-f007:**
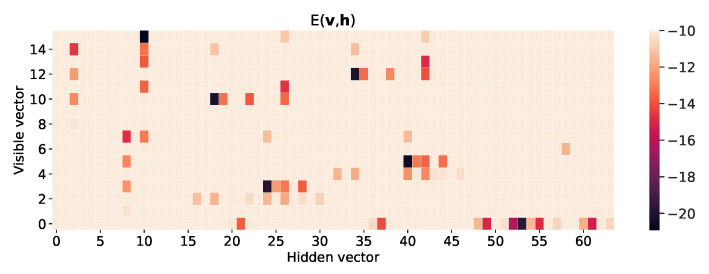
Heat map of energy function E(v,h;θ), representing the energy level of each configuration, after training of 2×2 bar-and-stripe patterns for 50000 epochs. The sizes of visible and hidden layers are N=4 and M=6, respectively. The learning rate is r=0.15 and the value of *k* in CD-*k* is 100. The vertical and the horizontal axes represent each configuration of the visible and the hidden layers, respectively. The black tiles represent the lowest energy configurations among all configurations, thus the probability of finding that configuration is high.

**Figure 8 entropy-22-00538-f008:**
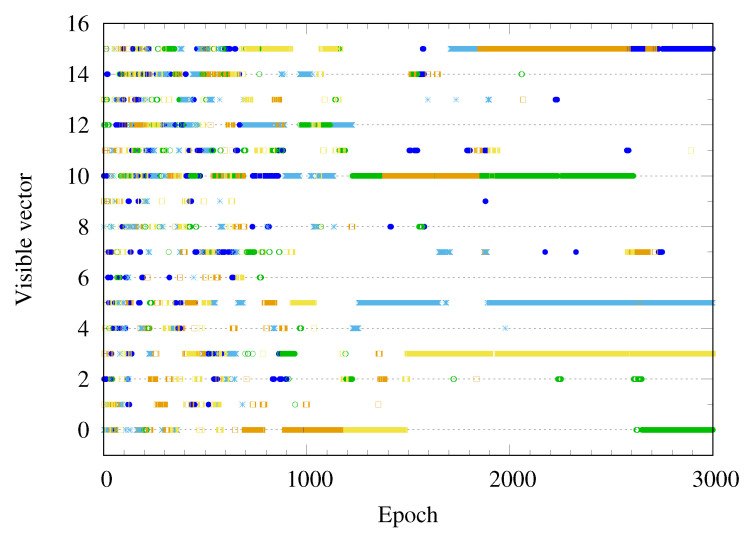
Markov chain Monte-Carlo trajectories of the visible vector $v_{i}$ are plotted as a function of the epoch. The visible vector jumps frequently in the early state of training and becomes trapped into one of the target states as the training proceeds.

**Figure 9 entropy-22-00538-f009:**
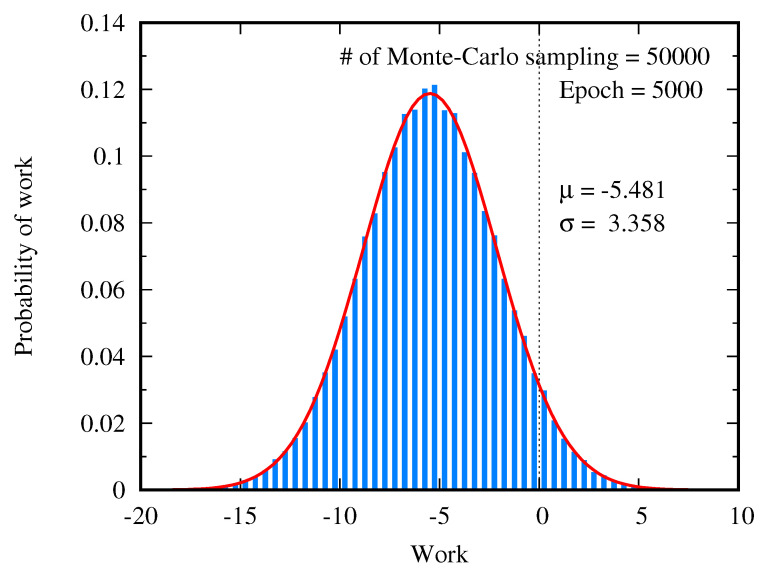
Gaussian distribution of work done by the restricted Boltzmann machine (RBM) during the training. The number of the Monte-Carlo sampling is 50000. The red curve is the plot of the Gaussian distribution using the mean and the standard deviation calculated by the Monte-Carlo simulation.

**Figure 10 entropy-22-00538-f010:**
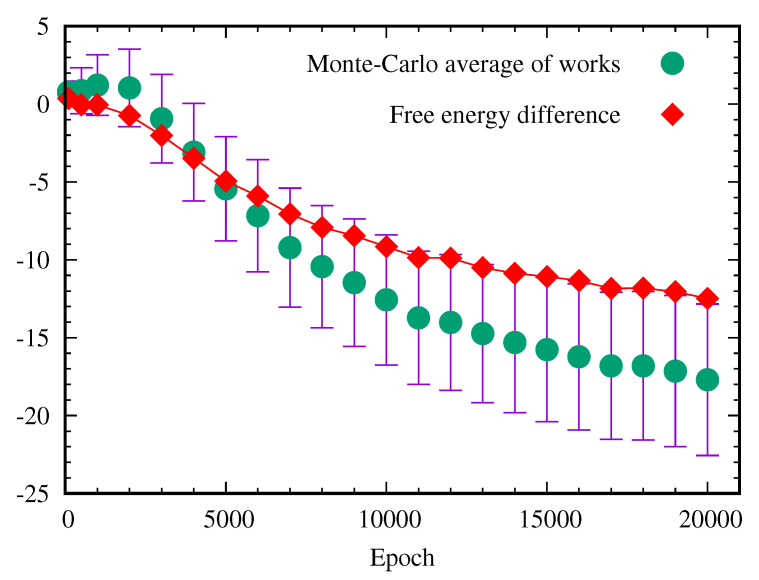
Average of work done with standard deviation and free energy difference ΔF=F(epoch)−F(epoch=0) as a function of the epoch. The error bar of the work represents the standard deviation of the Gaussian distribution.

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
