# Peer review of "Entropy, Free Energy, and Work of Restricted Boltzmann Machines"

_entropy, 2020, doi:10.3390/e22050538_

Round 1

Reviewer 1 Report

In this paper, the training process of the Restricted Boltzmann Machine (RBM) from the statistical physics perspective was investigated. The Authors analyzed how the thermodynamic quantities, like entropy, free energy, internal energy or work, change during the RBM learning. In the literature, the majority of papers concerning machine learning topics are based on the same schema: an algorithm is proposed and its superiority over other methods in terms of accuracy or other related measures is proved in a series of numerical experiments. There is a lack of works involving a deep theoretical insight into machine learning algorithms. Therefore, in my opinion, the reviewed paper is very important, unique, and valuable. Moreover, the reference list is very comprehensive and allows the readers to extend their knowledge about RBMS both from the computer science and physical point of view. Summarizing, I recommend accepting the paper after performing a minor revision according to the complaints presented below:

  1. In Introduction, it should be "... an undirected Markov random field" nstead of "... an undirected Markov random fields";
  2. In Introdution, it should be "... for example, in dimensionality reduction ..." instead of "... for example, dimensional reduction ...";
  3. In Introduction, it should be "... an RBM ..." or "... the RBM ..." instead of "... a RBM ...";
  4. In section 2.1, I noticed some little inconsistency: sometimes the visible unit called vi, and sometimes it is called the "visible unit i". I suggest to call in either visible unit vi or the i-th visible unit. The same applies to hidden units;
  5. In section 2.1, it would write "The joint probability of finding v and h of the RBM in a particular state is given by ..." instead of "The joint probability of finding v and h of the RBM is given by ...";
  6. In section 2.1, equation (2), it should be "Z(θ)" instead of "Z" in the denominator;
  7. In section 2.1, equation (4) and before it: why there is a probability distribution q(v)? Why not call it pdata(v)? Isn't it the same?;
  8. In section 2.1, I've never seen the name "constrast-divergence" for the CD algorithm. Maybe it would be better to write "Contrastive divergence" as it is usually called in the literature.;
  9. In section 2.2, between equations (8) and (9), shouldn't it be 0ln0=0 instead of 0ln0=1?;
  10. In section 2.2, it should be "The weights wij are ..." instead of "The weight wij are ...".;
  11. In section 2.2, caption of Fig. 4: I would write "... and k=100 in CD-k." instead of "... and CD-k 100.";
  12. In section 2.3, equation (17): in the second term it should be v2h2 after the arrow, instead of v1h1.;
  13. In section 2.3, caption of Fig. 7: It should be "... and the value of k in CDk is 100." instread of "... and the value of CDk is k=100."
  14. The Authors sometimes write CDk and sometimes CD-k. I recommend unifying this notation in the text;
  15. In section 2.3, it should be "So the main contribution ..." instead of "So the main contrition ...";

Finally, I have some open questions which I would like the Authors to refer to. Please, consider also including the answers to these questions in the revised version of the paper:

1. Why the size of the hidden layer was chosen to M=6?

2. In section 2.2, comment to Fig. 5: why the result "implies that the entropy may be another good indicator to monitor the progress of RBM while other thermodynamic quantities may be not"? Could you explain how this conclusion was derived?

3. In section 2.2, comment to Fig. 6: Could you explain how the following conclusion was derived: "This implies that the entropy of the visible layer, as well as the total entropy shownvin Fig. 5, can be an indicator to learning better than the reconstructed cross entropy C, Eq. (5)."? 

Reviewer 2 Report

Generally, I'm happy with the results of this manuscript. Mainly, suggesting that the entropy of the system can be a good indicator to the RBM progress along with Kullback-Leibler divergence.

Comment:

The authors used the Monte-Carlo calculation of the exponential average of work, to check the Jarzynski equality. Explain why. Is there any other way to check this?
